# Atomistic simulation of voltage activation of a truncated BK channel

**Zhiguang Jia, Jianhan Chen***

Department of Chemistry, University of Massachusetts, Amherst, United States

## eLife Assessment

This **valuable** study addresses the structural basis of voltage-activation of BK channels using atomistic simulations of several microseconds, to assess conformational changes that underlie both voltage-sensing and gating of the pore. The findings, including movement of specific charged residues, combined with the degree to which these movements are coupled to pore movements, provide a **solid** basis for understanding voltage-gating mechanisms in this class of channels. This paper will likely be of interest to ion channel biologists and biophysicists focused on voltage-dependent channel gating mechanisms.

## Abstract

Voltage-dependence gating of ion channels underlies numerous physiological and pathophysiological processes, and disruption of normal voltage gating is the cause of many channelopathies. Here, long timescale atomistic simulations were performed to directly probe voltage-induced gating transitions of the big potassium (BK) channels, where the voltage sensor domain (VSD) movement has been suggested to be distinct from that of canonical Kv channels but remains poorly understood. Using a Core-MT construct without the gating ring, multiple voltage activation transitions were observed at 750 mV, allowing detailed analysis of the activated state of BK VSD and key mechanistic features. Even though the S4 helix remains the principal voltage sensor in BK, its vertical displacement is only ~3 Å and accompanied by significant lateral movements. The nature of the predicted VSD movement is in strong agreement with recent Cryo-EM structural studies of mutant BK channels with constitutively activated VSD. Free energy analysis based on the predicted activation transition yielded a total gating charge of 0.44 $e$ per VSD, consistent with the experimental range of 0.48–0.65 $e$. We further show that the ability of modest physical movements with a small total gating charge to drive effective voltage gating of BK can be attributed to large gradients in the local electric field as reshaped by the protein. Furthermore, the S4 movement is coupled to the pore opening through a non-canonical pathway that involves the tightly packed S4-S5-S6 interface. These distinct mechanistic features may be relevant to voltage gating of other ion channels where VSDs are not domain-swapped with respect to the pore-gate domain.

**\*For correspondence:**
jianhanc@umass.edu

**Competing interest:** The authors declare that no competing interests exist.

## Introduction

Voltage-dependent gating of ion channels mediates numerous physiological and pathophysiological processes (*Hodgkin and Huxey, 1952*; *Bezanilla, 2000*; *Sigworth, 2003*; *MacKinnon, 2003*; *Swartz, 2008*), by regulating ion flows in response to membrane potential changes in excitable cells such as cardiac myocytes and smooth muscle cells in the lung and blood vessels (*Brayden and Nelson, 1992*; *Wellman and Nelson, 2003*; *Plüger et al., 2000*; *Nelson and Quayle, 1995*; *Tanaka et al., 1998*; *Pérez et al., 1999*). Disruption of normal voltage gating is involved in many cardiovascular and neurological diseases, including epilepsy, mental retardation, chronic pain, hypertension, arrhythmias, stroke, and ischemia (*Sokolov et al., 2007*; *Ackerman, 1998*; *Kaczorowski and Garcia, 1999*;

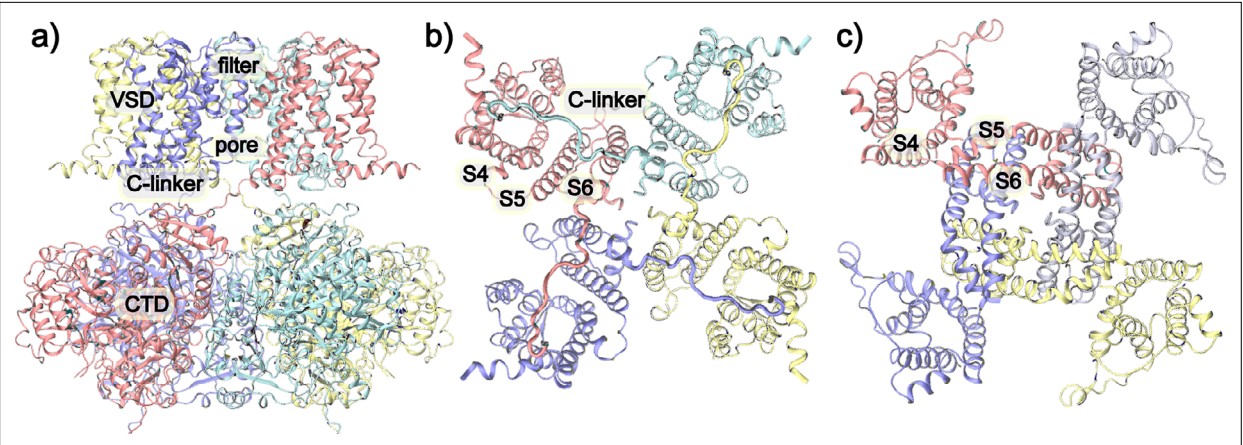

**Figure 1.** Overall structures of BK and Kv channels. (**a**) The overall structure of BK channels in the $Ca^{2+}$-free state (PDB: 6v3g) with key domains and regions labeled. Each monomer is shown in the same color. (**b**) Top view of the TMD of BK channels, showing the non-domain-swapped VSD/PGD configuration. (**c**) Top view of the TMD of Kv 1.2 channel (PDB: 3lut), where the VSDs and PGDs are domain-swapped. Note the much tighter packing of S4 from VSD and S5/S6 from PGD in BK channels.

*Lehmann-Horn and Jurkat-Rott, 1999*; *Cannon, 2006*; *Meldrum and Rogawski, 2007*; *Reid et al., 2009*; *Eijkelkamp et al., 2012*; *Bennett and Woods, 2014*). Among the family of potassium channels, the large-conductance potassium (BK) channels stand out in several ways (*Marty, 1981*; *Horrigan and Aldrich, 2002*; *Magleby, 2003*; *Salkoff et al., 2006*; *Contreras et al., 2013*; *Bentzen et al., 2014*; *Yang et al., 2015*). It has the largest single-channel conductance (up to ~300 ps) and is the only known $K^+$ channel to be activated by both intracellular $Ca^{2+}$ and membrane potential. Functional BK channels are homo-tetramers, with each subunit consisting of a seven-helix transmembrane domain (TMD) and a C-terminal $Ca^{2+}$-sensing cytosolic domain (CTD; *Figure 1a*). The deactivated state of BK channels contains a physically open central pore (*Hite et al., 2017*; *Tao et al., 2017*; *Tao and MacKinnon, 2019a*), lacking the classical bundle crossing constriction at the intracellular entrance observed in the voltage-dependent potassium (Kv) channels (*Long et al., 2005*). Instead, BK channels likely follow the hydrophobic gating mechanism, where the pore undergoes hydrophobic dewetting transition to create a vapor barrier to block ion permeation in the deactivated state (*Jia et al., 2018*; *Gu and de Groot, 2023*; *Nordquist et al., 2023*; *Deng and Cui, 2024*; *Coronel et al., 2024*). Furthermore, the voltage sensor domains (VSDs) of BK channels are not domain-swapped with respect to the pore-gate domain (PGD), in contrast to the domain-swapped TMD organization of Kv channels (*Figure 1b–c*). The difference in TMD organization is likely a key factor underlying important but still poorly understood differences in the voltage sensing and gating mechanisms of BK channels (*Zhou et al., 2017*; *Gustavo et al., 2023*; *Carrasquel-Ursula et al., 2022*; *Sun and Horrigan, 2022*; *Kallure et al., 2023*) in comparison to the 'canonical' framework established mainly through the study of Kv channels (*Larsson et al., 1996*; *Yifrach and MacKinnon, 2002*; *Schmidt et al., 2006*; *Islas and Sigworth, 1999*; *Catterall, 2010*; *Catterall and Zheng, 2015*; *Wu et al., 2016*; *Lenaeus et al., 2017*; *Matthies et al., 2018*).

In Kv channels, the linker connecting transmembrane (TM) helices S4 and S5 forms a α-helix (~15 aa) that wraps around the S6, with the S4 of one subunit interacting with the S5 of a neighboring subunit and simultaneously connecting to the S5/S6 within the same subunit through the S4-5 linker. Conversely, in BK channels, the S4-S5 linker is a short loop (~5 aa), and the S4 solely interacts with S5 within the same subunit (*Figure 1*). Within the canonical voltage activation of Kv channels, each of the first four arginine residues in the VSD S4 accounts for about 1e gating charge (*Schoppa et al., 1992*; *Aggarwal and MacKinnon, 1996*; *Long et al., 2007*; *Seoh et al., 1996*), giving rise to a total gating charge of ~3–3.5 e per monomer. Membrane depolarization drives S4 to move upward by ~8 Å, which causes the S4-S5 linker helix to pivot upwards and change the interactions with S6 and releases the S6 helix bundle crossing to physically open the gate (*Fowler and Sansom, 2013*; *Jensen et al., 2012*; *Hou et al., 2020*). For BK channels, despite sharing many conserved charged residues, the total number of gating charges is only 0.6 e per VSD (*Horrigan and Aldrich, 2002*; *Horrigan and Aldrich, 1999*; *Lorenzo-Ceballos et al., 2019*; *Carrasquel-Ursulaez et al., 2015*; *Contreras et al., 2012*;

*Stefani et al., 1997*). In addition to arginine residues on BK S4 (R207, R210, and R213), other residues also appear to contribute to voltage sensing, such as D153 and R167 on S2, D186 on S3, and E219 on S4 (*Carrasquel-Ursula et al., 2022*; *Ma et al., 2006*; *Díaz et al., 1998*). Furthermore, fluorophore quenching suggested that the movement of BK VSDs during voltage gating was smaller and involved both vertical and lateral motions (*Pantazis et al., 2010*; *Pantazis and Olcese, 2012*; *Savalli et al., 2006*). At present, there remains significant ambiguity in the identity of gating charges, details of VSD motions, and how they drive the pore opening in BK channels.

In this study, extensive atomistic molecular dynamic (MD) simulations up to 10 μs in length were performed in explicit solvent and membrane to directly probe the voltage-driven activation of BK channels in the $Ca^{2+}$-free state, using the Core-MT construct that does not include the CTD but retains voltage gating (*Budelli et al., 2013*; *Zhang et al., 2017*). We were able to directly observe spontaneous opening transitions and ion conductance of the channel under 750 mV membrane voltage within the 10 μs simulation timescale. The observed VSD movements appear to be highly consistent with two recent Cryo-EM structural studies of mutant BK channels with constitutively activated VSDs (*Gustavo et al., 2023*; *Kallure et al., 2023*). Free energy analysis was performed to further quantify the total gating charge as well as the contributions of key charged residues. The results are in quantitative agreement with available experimental data (*Carrasquel-Ursula et al., 2022*; *Ma et al., 2006*; *Díaz et al., 1998*), providing further support for the predicted voltage sensing and gating mechanism. The analysis reveals central roles of voltage-induced displacement of R210 and R213 on S4 in voltage sensing and the strong interactions at the S4-S5-S6 interface in VSD-pore coupling. This mechanism was further validated using steered MD simulations showing that pulling on the charged tip of R210 and R213 alone could readily drive pore opening in Core-MT BK channels. Taken together, the current work provides for the first time a reliable detailed molecular mechanism of BK voltage activation, which is distinct from the canonical voltage gating mechanism of Kv channels. These novel voltage gating principles are likely relevant in understanding the gating and regulation of other non-domain swapping ion channels.

## Results and discussion
### Direct atomistic simulations of voltage activation of BK channels

Starting from a fully equilibrated closed state, multiple atomistic simulations were performed at 0 and 750 mV membrane voltages for up to 10 μs to directly probe voltage-driven activation of Core-MT BK channels using the special purposed supercomputer Anton 2 (*Shaw et al., 2008*; *Shaw et al., 2014*; see Materials and methods; *Figure 2—figure supplement 1*). The simulated voltage of 750 mV is higher than the experimental conditions, where $V_{1/2}$ of Core-MT BK channels is ~244 mV compared to ~180 mV for the full-length BK channel (*Zhang et al., 2017*), to accelerate protein conformational transitions. Similar voltages have been used in atomistic simulations and found to generate realistic transitions (*Jensen et al., 2012*; *Jiang et al., 2017*). We compare the membrane thickness at 300 and 750 mV and the results reveal no significant difference in the membrane thickness (*Figure 2—figure supplement 2*). At 0 mV, the Core-MT BK channel remained highly stable in an apparently closed state, with the pore fully dehydrated and not permeable to ions (*Figure 2—figure supplement 1*, top row). Similar to what has been observed previously in simulations and cryo-EM maps (*Tao and MacKinnon, 2019a*; *Jia et al., 2018*; *Gu and de Groot, 2023*; *Nordquist et al., 2023*; *Deng and Cui, 2024*; *Gustavo et al., 2023*), lipid tails can enter the dewetted pore through the fenestration gap between the pore lining S6 helices. They are highly dynamic and likely contribute to the stability of the dewetted state of the pore. Note that the absence of CTDs increases the flexibility of pore-lining S6 helices, and the fully relaxed pore profile (red trace in *Figure 2—figure supplement 1d, top row*) shows substantial differences compared to that of the $Ca^{2+}$-free Cryo-EM structure of the full-length channel (black trace). Importantly, the VSDs were highly stable and showed minimal movements (~1 Å or less) for all charged groups as well as the TM helices themselves.

At 750 mV, significant movements were observed with multiple charged groups on S4 (*Figure 2a*, *Figure 2—figure supplement 1*). In particular, the guanidium groups of R210 and R213 moved upward along the membrane normal (z-axis) by up to ~8 and~10 Å, respectively, plateauing after ~5 μs (e.g. see *Figure 2a*). Other VSD charges showed much smaller z-displacements. The apparent modest movements of ~3–4 Å of E219 and R207 seem to be mainly a result of the upshift of S4 itself,

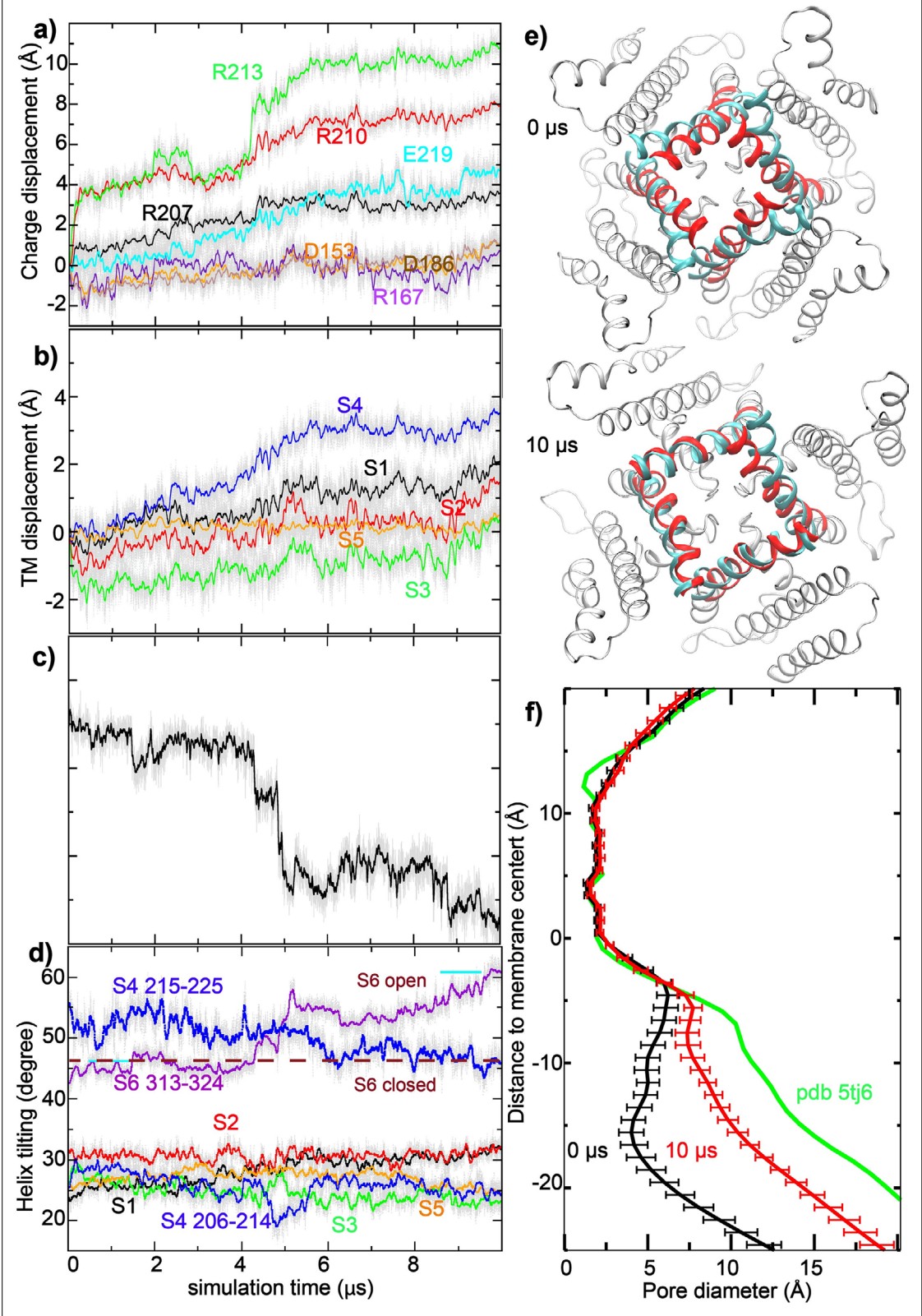

**Figure 2.** Voltage activation of Core-MT BK channels. (**a–d**) Results from a 10-µs simulation under 750 mV (*sim2b* in *Supplementary file 1*). Each data point represents the average of four subunits for a given snapshot (thin gray lines), and the colored thick lines plot the running average. (**a**) z-displacement of key side chain charged groups from initial positions, (**b**) z-displacement of centers-of-mass of VSD helices from initial positions, (**c**) backbone RMSD of the pore-lining S6 (F307-L325) to the open state, and (**d**) tilt angles of all TM helices. The locations of charged groups were

*Figure 2 continued on next page*

*Figure 2 continued*

taken as those of guanidinium CZ atoms (for Arg) and side chain carboxyl carbons (for Asp/Glu). Only residues 313–324 of S6 were included in tilt angle calculation, and the values in the open and closed Cryo-EM structures are marked using purple dashed lines for reference in panel d. (**e**) Superimposition of the initial (0 μs) and final (10 μs) structures of the pore (residues F307–L325; red cartoon), in comparison to the open Cryo-EM structure (cyan cartoon). The view shown is from the bottom (cytosolic side). (**f**) Average pore profiles calculated from the first and last 0.1 μs of *sim2b*, with error bars showing standard error. The pore profile derived from PDB 5tj6 (open state) is shown as a reference.

The online version of this article includes the following figure supplement(s) for figure 2:

**Figure supplement 1.** The charged group z-displacement, z-displacement of the centers of mass of TM helices, number of pore waters, the averaged pore profiles during four Anton 2 simulations of Core-MT BK channels at 0 mV (row 1), 750 mV (rows 2–3), and 300 mV (rows 4–6) membrane voltages.

**Figure supplement 2.** Distribution of lipid phosphor atoms under different membrane voltages.

which reached a maximum of ~3 Å during the second half of the simulation (*Figure 2b*). In contrast, other VSD helices S1-S3 exhibited much smaller z-axis movements (~1 Å). As illustrated in *Video 1*, the large charge displacements of R210 and R213 charges with modest S4 movement were enabled by side chain snorkeling.

Voltage-driven VSD movements were apparently coupled with an opening transition of the pore, allowing one to directly observe Core-MT activation and $K^+$ conductance during both 10-μs atomistic simulations on Anton 2 (*Figure 2—figure supplement 1*). As illustrated in *Figure 2* and *Video 2*, the backbone root-mean-squared distance (RMSD) of the pore from the open state decreased sharply from ~4.5 Å around the 4 μs mark to below ~2.7 Å around the 5 μs mark, as the S4 helix shifted up along the z-axis by ~2 Å during the same time span. The pore opening transition mainly involved the increase of the tilt of S6 segments below the glycine hinge (residues 313–324; *Figure 2d*). The pore continued to move closer to the open state after the initial rapid response to VSD activation, eventually reaching a state that has similar S6 helix tilt and is only ~2.2 Å from the $Ca^{2+}$-bound Cryo-EM structure at the end of the 10 μs run (*Figure 2e*, red vs. cyan cartoons). These pore structural changes roughly doubled the pore diameter, from ~10 Å to ~20 Å at the intracellular entrance (z ~ − 20 Å; *Figure 2f*). It should be noted that the full-length BK channel in the $Ca^{2+}$-bound state has an even larger intracellular opening (*Figure 2f*, green trace), suggesting that additional dilation of the pore may occur at longer timescales, or in response to Ca-binding to the full-length channel. On the other hand, the simulation construct does not include the 11-residue Kv mini-tails required for assembly and membrane insertion of Core-MT (*Budelli et al., 2013*), which could impact the pore conformation. Furthermore, the single-channel conductance of Core-MT is ~30% lower (*Budelli et al., 2013*), suggesting that its open pore is more constrictive than that of the full-length BK channels.

The opening transitions were accompanied by rehydration of the pore and the channel became conductive to $K^+$ (*Figure 3* and *Video 3*). Consistent with the smaller pore diameter, the final states of the pore in the simulations only accommodate ~30 waters, compared to ~40 waters for the $Ca^{2+}$-bound state of full-length BK (*Jia et al., 2018*). Furthermore, the single-channel conductance estimated from the last 2 μs of *sim 2b* (*Figure 3a*) is

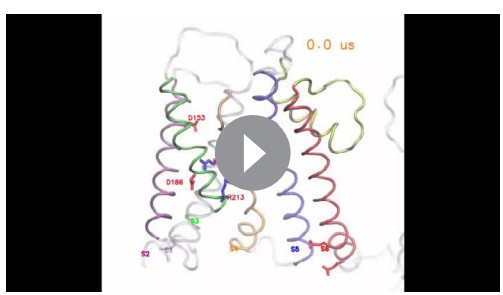

**Video 1.** Movement of gating charges and TM helices S1-S6 during voltage activation of Core-MT BK channel (sim 2b). Key charges R210 (S4) and R213 (S4) are colored in blue, D153 (S2) and D182 (S3) in red. In addition, conserved S6 charges, E321 and E324, at the cytosolic entrance of the pore are shown in red sticks.
https://elifesciences.org/articles/105895/figures#video1

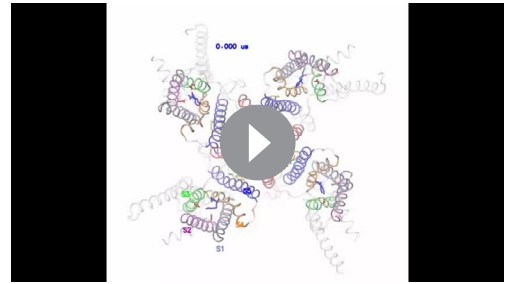

**Video 2.** Movement of pore lining S6 helices (red cartoon) during voltage activation of Core-MT BK channel (sim 2b). The view is from the bottom (cytosolic side). The arrangement of S6 helices in the Ca2+-bound state of full-length BK channel is shown in yellow cartoon for reference. E321 and E324 at the cytosolic entrance of the pore are shown in green sticks.
https://elifesciences.org/articles/105895/figures#video2

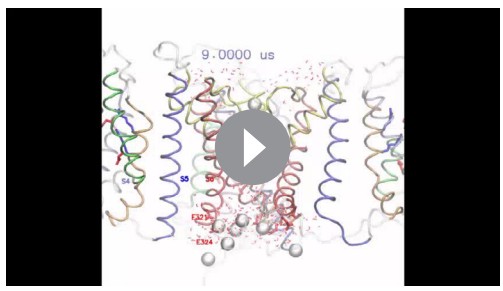

**Figure 3.** Pore rehydration and ion conductance. (**a**) The number of water molecules inside the pore as a function of time during simulation (*sim 2b*), with the upper panel showing recorded ion permeation events. Inserts show snapshots of the pore region at three representative time points. (**b**) Snapshots illustrating key steps of K⁺ ions passing through the filter. Potassium ions inside or near the filter are colored according to their identities. The water molecule bridging two ions inside the filter is also shown as van der Waals spheres.

The online version of this article includes the following figure supplement(s) for figure 3:

**Figure supplement 1.** Conductance of the open state of Core-MT derived from the Ca²⁺-bound full-length BK structure (PDB: 5tj6).

**Video 3.** K⁺ permeation events during the last 1 us of sim 2b, when the dilated pore is hydrated and conductive. The K+ions are presented as spheres and colored by the atom index. Water molecules within the pore region are shown using red sticks.

https://elifesciences.org/articles/105895/figures#video3

only ~1.5 pS, much lower than the experimental value of ~220 pS for Core-MT (*Zhang et al., 2017*). We note that classical force fields, such as CHARMM36m used in the current simulations, are known to underestimate the single channel conductance by about one order of magnitude (*Furini and Domene, 2020*). Indeed, the fully opened state of Core-MT, constructed from the Ca²⁺-bound Cryo-EM full-length BK structure, is predicted to have a conductance of ~6 pS using the same simulation setup (*sim 7*; *Figure 3— figure supplement 1*), which is only ~4 fold of that of the final state from the voltage activation simulations. We further note that subconductance open states have been observed in single channel

recordings of BK channels (*Ferguson et al., 1993*; *Sun et al., 1999*; *Gonzalez-Perez et al., 2012*; *Takacs et al., 2014*). Besides the limitation of the current fixed charge force fields in quantitatively predicting channel conductance, we note that the molecular basis for the large conductance of BK channels is actually poorly understood (*Mironenko et al., 2021*). It is noteworthy that the pore hydration level appears to be an important factor in determining the apparent conductance in the simulation, which has also been proposed in a previous atomistic simulation study of the *Aplysia* BK channel (*Gu and de Groot, 2023*). Despite not reaching a fully conductive state, the ability of the dilated and hydrated pore to sustain K⁺ permeation throughout the second half of the 10 µs simulation demonstrates that the hydrophobic gate in the closed channel has been broken and the channel enters a conductive state. Consistent with previous simulations of other K⁺ channels (*Mironenko et al., 2021*; *Bernèche and Roux, 2001*; *Roux, 2005*), the conductance follows a multi-ion mechanism, where there are at least two K⁺ ions inside the filter, preferentially occupying positions S1/S3 or S2/S3 (*Figure 3b*). The two bound ions subsequently move to S0/S1 when an incoming ion takes the S3 position. Notably, between the ions inside the filter, there can be either one or no water molecule, indicating the coexistence of both soft and hard knock-on mechanisms.

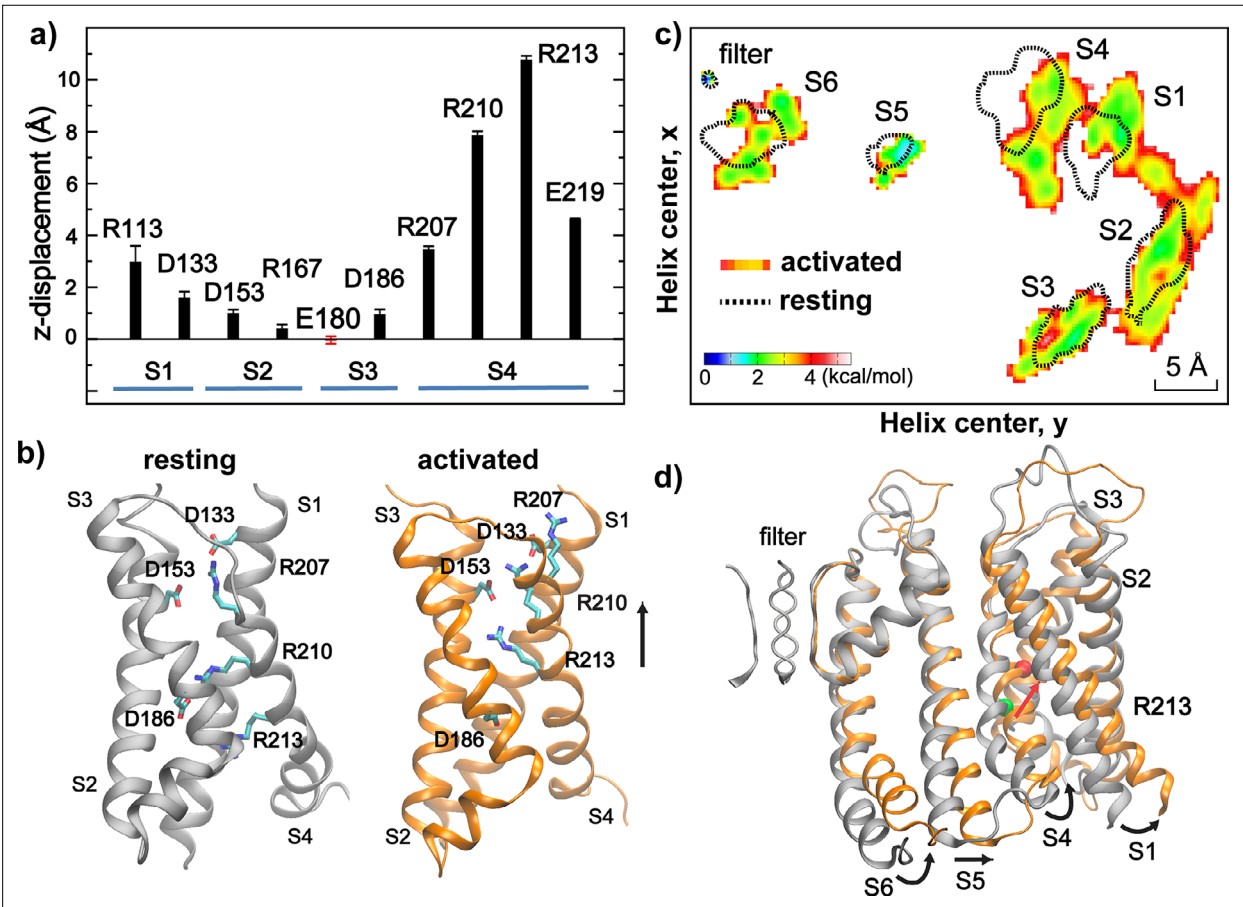

**Figure 4.** VSD gating charge and voltage-sensing movements. (**a**) Average voltage-induced movements of key charges along the membrane normal (z-axis) with respect to the initial resting state structure, derived from last 500 ns of 750 mV simulation *sim 2b*. Error bars show the standard deviations. (**b**) Conformations of key charged residues in the resting (silver) and activated (orange) states of BK VSD. The resting and activated states are represented using the snapshots at 0 and 10 µs of sim2b, respectively. (**c**) Distributions of centers of mass of TM helices along the membrane lateral directions (**x and y**) (view from the cytosolic side). The distributions for resting and activated states were derived from the first and last 500 ns of the 750 mV simulation *sim 2b*, respectively, which were converted into the free energy scale by ~R T ln *P(x,y)* with T=300 K. The contour for the resting state distribution (dotted lines) is drawn at 4 kcal/mol. (**d**) Overlay of the resting (silver) and activated (orange) states of the TM domain of the Core-MT BK channel. The green and red spheres mark the backbone Cα atom of S4 R213 in the resting and activated states. Only one subunit is shown for clarity, but all four filter loops are shown for reference.

The online version of this article includes the following figure supplement(s) for figure 4:

**Figure supplement 1.** Steered MD simulations of BK activation.

## Gating charges: how small VSD movements support voltage sensing

The ability of Anton 2 simulations to directly observe voltage-driven activation of Core-MT BK channels allows one to examine the molecular details of VSD activation and VSD-pore coupling. An important observation is that, even though the charged groups of R210 and R213 show z-displacement of 8–10 Å (*Figure 4a*), the overall movement of S4 along the membrane normal is only ~3 Å, less than a single α-helical turn (*Figure 2b*). This is considerably smaller than that observed in canonical Kv channels, which has been estimated to be ~8 Å (*Fowler and Sansom, 2013*) and up to 15 Å (*Jensen et al., 2012*). Smaller S4 movements have been suggested in fluorophore quenching experiments (*Pantazis et al., 2010*; *Pantazis and Olcese, 2012*; *Savalli et al., 2006*) and molecular simulations (*Carrasquel-Ursula et al., 2022*). Recent Cryo-EM structures of mutant BK channels with constitutively activated VSDs actually reveal minimal S4 movements along the membrane normal (*Gustavo et al., 2023*; *Kallure et al., 2023*). Upon voltage activation, R210 switches the salt-bridge partner from D186 on S3 to D153 on S2 and D133 on S1, R213 switches to engage with D153, and R207 switches from engaging with D153 and D133 to becoming exposed to the water/membrane interface (*Figure 4b*). Similar movements were also observed in recent high-resolution Cryo-EM structures of R207A mutant BK channels with constitutively activated VSDs at 0 mV (*Kallure et al., 2023*), even though the net z-displacements of R210 and R213 charges in Cryo-EM structures are about ~2–4 Å smaller. Both the smaller side chain charge movements and a lack of overall S4 z-displacement in the mutant Cryo-EM structures may be due to the absence of membrane voltage. To further evaluate if z-placements observed at 750 mV are an artifact of unphysical voltage, three independent simulations were initiated from the final state of *sim 2b* at 300 mV for 1.0 μs (*sim 9* in *Supplementary file 1*). The results, summarized in *Figure 2—figure supplement 1*, show that the activated state of VSD remained stable and the pore stably hydrated in all three simulations.

Another important feature is that multiple helices in VSD underwent substantial outward movements in the membrane lateral direction in addition to the modest z-displacement of S4. As shown in *Figure 4c*, both S1 and S4 displayed ~3 Å outward movements within the x-y plane. These lateral movements are apparently important for driving the opening motion of the pore-lining S6 helices (*Figure 4d*), likely mediated by direct S4-S5-S6 interactions (see the next section). Importantly, the predicted VSD movements appear highly consistent with observations from fluorophore quenching experiments (*Pantazis et al., 2010*; *Pantazis and Olcese, 2012*; *Savalli et al., 2006*), which suggest that S4 likely moves towards S3 and closer to S1/S2 upon activation. Interestingly, the simulations predict that S1 moves during voltage activation instead of S2, in contrast to the previously proposed model (*Pantazis and Olcese, 2012*). Lateral movements of S4 at the cytosolic end have also been observed in recent Cryo-EM structures of R202Q mutant *Aplysia* BK channel (equivalent to R213Q in human BK), which presumably locks the VSDs in the activated state (*Gustavo et al., 2023*), even though the overall movements of VSD are much more subtle in the Cryo-EM structures. The latter may again be a direct consequence of the absence of membrane voltage.

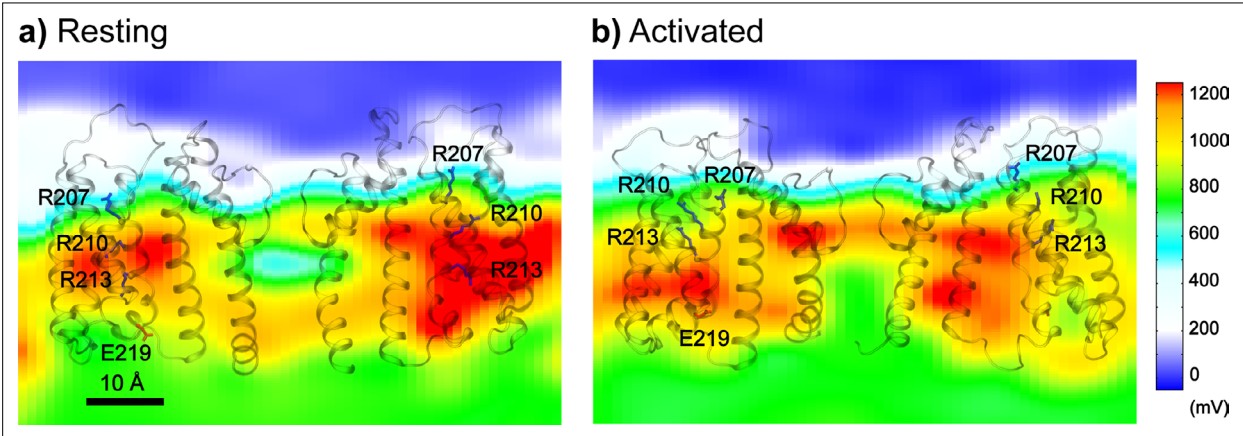

**Figure 5.** Electrostatic potential fields of the Core-MT BK channel at 750 mV with resting and activated VSDs. The fields were calculated as the averages of the first (resting) and last (activated) 250 ns of simulation *sim2b*. The fields are shown on a plane that goes through the filter and R210. Only two subunits of Core-MT BK are shown for clarity, with side chains of key S4 charges shown in sticks.

**Table 1.** Residue contributions to the gating charge per VSD.

| Residues | Current work* | Carrasquel-Ursula et al., 2022 | Ma et al., 2006 | Díaz et al., 1998 |
|---|---|---|---|---|
| D153 (S2) | 0.08±0.04 | –0.03 | 0.26 | |
| R167 (S2) | 0.002±0.03 | –0.02 | 0.15 | |
| D186 (S3) | –0.01±0.03 | 0.0 | 0.22 | |
| R207 (S4) | 0.04±0.01 | –0.01 | –0.03 | 0.275 |
| R210 (S4) | 0.25±0.02 | 0.32–0.35† | –0.01 | 0.325 |
| R213 (S4) | 0.19±0.01 | 0.23–0.31† | 0.34 | 0.3 |
| E219 (S4) | –0.09±0.02 | 0.03 | 0.02 | |

*Statistical error was determined by block averaging.
†Results from reversed charge mutations are excluded here.

To understand how the modest movements of VSDs can support effective voltage sensing in BK channels, we calculated the average electrostatic potential maps for the resting and activated states (*Figure 5*). The results reveal how the protein significantly remodels the local electric field, creating large gradients along both the membrane normal and lateral directions. This is similar to the electric field focusing effects proposed for VSDs of both Kv and BK channels (*Carrasquel-Ursula et al., 2022*; *Starace and Bezanilla, 2004*). In particular, residues R210 and R213 reside in a region of high electrostatic potential (>1100 mV) in the resting state, but move to a region of much lower electrostatic potential (~600 mV) in the activated state. Therefore, despite the modest S4 movements, charges on R210 and R213 can effectively sense a voltage change of ~500 mV, compared to the total imposed membrane voltage of 750 mV.

We performed free energy calculations (*Khalili-Araghi et al., 2010*; *Roux, 2008*) to further quantify the contributions of key VSD residues to the total gating charge of BK channels (see Materials and methods; *Supplementary file 1 sim 3–6*). As summarized in *Table 1*, the calculations estimated the total gating charge per VSD to be ~0.45 *e*, in strong agreement with the range of 0.48–0.65 *e* measured experimentally for the full-length BK channel (*Carrasquel-Ursula et al., 2022*; *Ma et al., 2006*; *Díaz et al., 1998*). The results further identify R210 and R213 as the primary contributors to voltage sensing, accounting for approximately 97% of the total gating charge per VSD (*Table 1*). The calculated residue contributions to gating charge are highly consistent with the latest experimental measurements by *Carrasquel-Ursula et al., 2022*, even though earlier experimental studies disagreed on the contributions of several residues, including R207, R167, D153, and D186. It has been recognized that interpretation of gating charge measurements on mutant BK channels is nontrivial (*Carrasquel-Ursula et al., 2022*). For example, neutralizing R213 reduces the total gating charge from 2.62 *e* for WT to 1.31 *e* for R213C; yet replacing it with a negative charge (R213E) does not further reduce the net gating charge (*Ma et al., 2006*). The simulations reveal that the complication in experimental analysis may be attributed to modest movements of VSD in both membrane normal and lateral directions and a clear interplay of VSD conformation and local electric field (*Figure 5*). Nonetheless, atomistic simulations together with electrostatic potential and free analyses strongly support a central role of R210 and R213 in BK voltage sensing.

The dominant role of R210 and R213 in voltage sensing of BK channels was further tested by steered MD simulations (*sim 8*), where biasing potentials were applied to steer the movement of the guanidium CZ atoms of R210 and R213 from their positions in the initial resting state to those in the activated state in absence of membrane voltage (see Materials and methods and *Figure 4—figure supplement 1*). The simulations show that upward movements of R210 and R213 guanidium tips alone can readily drive pore opening, which was observed in two out of four independent steer MD simulations (*Figure 4—figure supplement 1*). In the other two simulations, the S5-S6 packing was ruptured during the first 100 ns, likely due to the rapid rate of pulling. Similar to simulations under 750 mV membrane potential, pore dilation and increase in S6 tilting were observed as the VSD responded to the movement of R210 and R213, along with the breakdown of the hydrophobic barrier, as indicated by an increase in the number of water molecules within the pore. The ability of R210 and R213 charge

upward movement alone to drive pore opening in the absence of membrane voltage further supports the dominant roles of these two charges in voltage sensing and gating of BK channels.

## Central role of the S4-S5-S6 interface in VSD-pore coupling

Examination of the covariance matrices (*Figure 6—figure supplement 1*) reveals that all TM helices, particularly S1-S5, are tightly coupled in both resting and activated states. The lateral movements of S1 and S4 during activation mainly involve the cytosolic ends, concerted with those of S5 and S6 from the pore (*Figure 4b*). In the Cryo-EM structures, a kink is observed in S4 below R213, resulting in approximately a~30° bend in the C-terminal half away from S5 and the pore. During voltage-induced activation, the N-terminal region of S5 maintained strong contacts with the S4 C-terminal segment (*Figure 6—figure supplement 2*), such that the upward and outward movement of S4 induces an increase in the tilt and bend of S5 (*Figure 4d*). This S4-S5 movement, in turn, increases the tilt angle of S6 C-terminal half (F315 to E324) by ~15° to dilate the pore. In particular, the hydrophobic helix-helix contacts between S5 (L235, L239, and F242) and S6 (F315, V319, and I322) are strengthened, while those between N231 and K234 on S5 and E321 and E324 on S6 are weakened during this process (*Figure 6—figure supplement 2*). Besides pore dilation, the conformational response of S6 alters the orientation of the conserved E321 and E324 at the cytosolic entrance from membrane interface-facing in the closed state to pore-facing in the open state, as observed in Cryo-EM structures (*Hite et al., 2017*; *Tao et al., 2017*; *Tao and MacKinnon, 2019a*). The resulting increase in both pore size and surface hydrophilicity promotes pore hydration and disrupts vapor barrier to render the channel conductive.

To further identify structural elements central to BK VSD-pore coupling, we performed unbiased dynamic community and coupling pathway analysis based on the covariance matrices derived from atomistic simulations (see Materials and methods). The results, summarized in *Figure 6a*, reveal that S4-6 forms a single community distinct from the rest of the VSD (and S0 helix), where the motions of residues are more strongly coupled to each other than to the rest of the channel. Indeed, the optimal and suboptimal pathways of allosteric coupling between S4 R213 (a major voltage sensing charge) and S6 E321 (a key position of pore dilation and hydrophilicity increase) pass exclusively through S5 residues (*Figure 6b*). The central role of S5 in mediating VSD-pore coupling is further supported by analyzing the information flow betweenness (*Botello-Smith and Luo, 2019*), which provides a global measure of how conformational perturbation flows through the network from the 'source' (e.g. R213) to the 'sink' (e.g., E321). The analysis reveals that S5 residues have the largest contributions to the informational flow besides neighboring residues on S4 or S6 (*Figure 6c*). Other residues on S1 or S3 also contribute significantly to the flow, albeit at much lower levels compared to S5. Interestingly, analysis of the simulation trajectory of $Ca^{2+}$ fully open state (*Supplementary file 1*, *sim 7*) reveals highly similar patterns in the dynamic coupling communities, coupling pathways as well as informational flow (*Figure 6—figure supplement 3*), despite substantial differences in the VSD conformation and S4-S5-S6 packing (*Figure 6—figure supplement 4*). The implication is that $Ca^{2+}$- and voltage activation pathways likely have significant overlaps, which is consistent with experimental observations that BK channels can be synergistically or independently activated by membrane depolarization and $Ca^{2+}$ binding. Taken together, the simulation and dynamic network analysis consistently point to a central role of the S4-S5-S6 interface in VSD-pore coupling of BK channels.

## Concluding discussion

Long atomistic simulations in explicit water and membrane performed using special-purpose super-computer Anton 2 have allowed direct observation of voltage-induced activation of the Core-MT BK channels for the first time. These simulations provide crucial new insights into how BK VSDs may sense membrane voltage and how the VSD movements may drive the pore opening and activate channel conductance. In particular, the simulations support that the S4 helix remains the major voltage-sensing element and that it undergoes modest movements of ~3 Å along both the membrane normal and lateral directions. The ability of BK channels to utilize relatively small VSD movements to sense membrane voltage and drive channel activation is likely attributed to the protein's ability to modify the local electric field, which leads to large electrostatic potential gradient in both vertical and lateral directions. Further analyses reveal that R210 and R213 are the major voltage sensing residues in the wild-type Core-MT BK channel, contributing ~97% to the total gating charge of ~0.45 *e* per

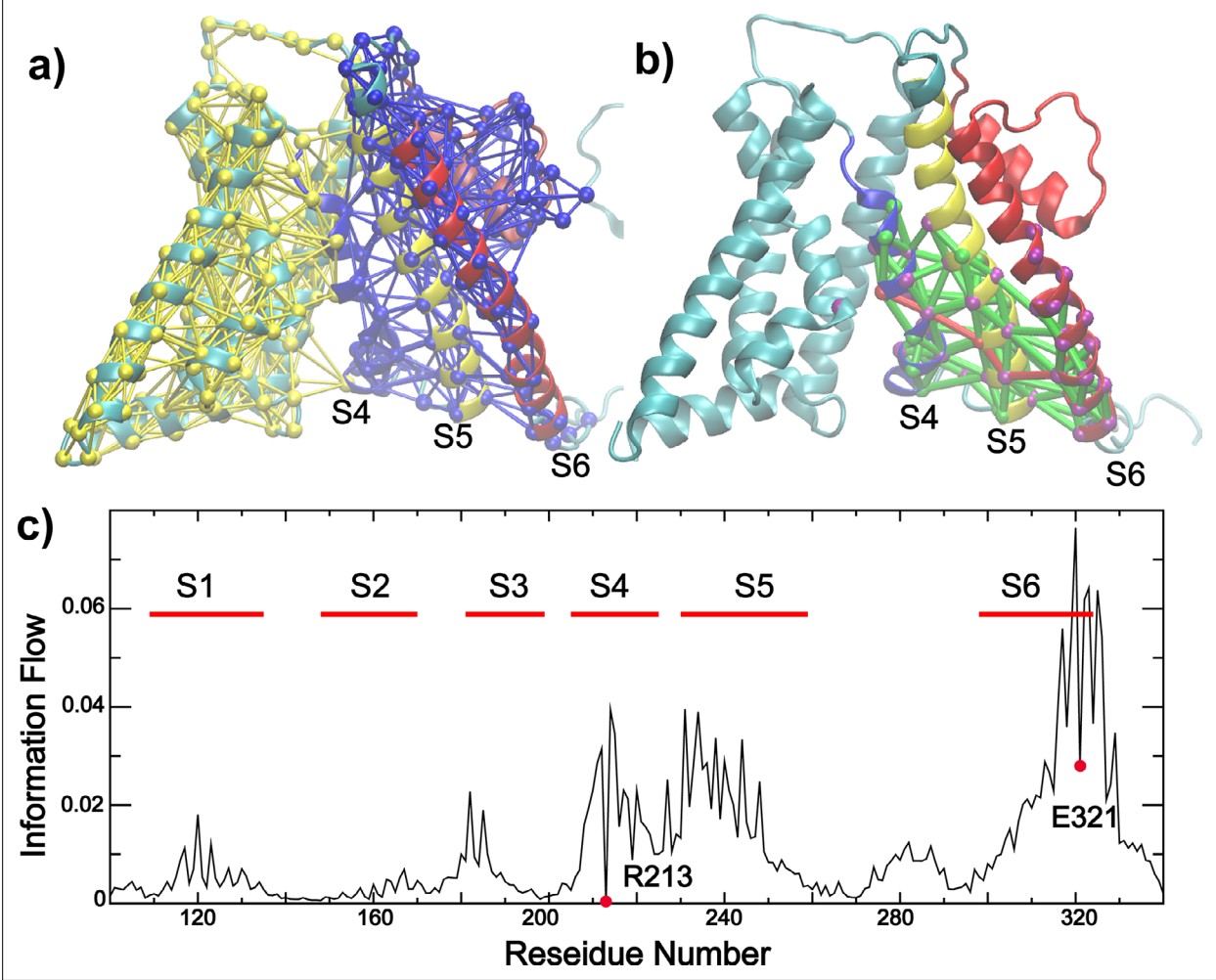

**Figure 6.** Dynamic community, coupling pathways, and information flow of VSD-pore coupling in BK. (**a**) Dynamic community analysis showing that TM S4-6 are clustered into single tightly coupled community (blue network). The nodes (residues) and edges (contacts) are colored based on the community number. (**b**) Optimal and suboptimal pathways of dynamic coupling between R213 (VSD S4) and E321 (pore-lining S6). All paths are colored green except for the optimal path, which is colored red. Nodes with information flow value >0.02 are highlighted in purple; (**c**) Information flow profile of the Core-MT BK channel with R213 as the source and E321 as the sink node (labeled by red circle), respectively. All dynamic coupling analysis was derived from the last 500 ns of *sim 1* (closed state at 0 mV; see ***Supplementary file 1***).

The online version of this article includes the following figure supplement(s) for figure 6:

**Figure supplement 1.** Covariance matrices of the Core-MT BK channel before (left) and after (right) voltage-induced activation (sim 2b).

**Figure supplement 2.** Locking and concerted movements of S4, S5, and S6.

**Figure supplement 3.** Dynamic community, coupling pathways, and information flow of VSD-pore coupling derived from simulation of Ca²⁺-bound structure.

**Figure supplement 4.** Changes in residue-residue contact probability relative to the closed channel.

VSD. These core features of BK voltage sensing from simulation are largely consistent with a range of existing functional, biophysical, and structural studies (*Gustavo et al., 2023*; *Carrasquel-Ursula et al., 2022*; *Kallure et al., 2023*; *Pantazis and Olcese, 2012*), even though structural studies of mutant BK channels with constitutively activated VSDs reveal smaller movements of R210 and R213 guanidinium moieties and minimal vertical movement of S4. The later observations may be a consequence of the complete absence of membrane voltage during structural determination. Importantly, control simulations show that the activated VSD state remains stable at 300 mV, suggesting that the observed vertical S4 movement is not an artifact of high voltage of 750 mV.

Another distinct feature derived from atomistic simulation is that the tightly packed S4-S5-S6 interface in BK channels plays a central role in the VSD-pore coupling, which is further supported by

dynamic community, coupling pathway, and informational flow analyses. The modest movements in S4 drive a series of concerted conformational changes in S5 and eventually in S6, with the S4-S5 linker not playing a major role as observed in the canonical Kv channels (*Fowler and Sansom, 2013*; *Jensen et al., 2012*; *Hou et al., 2020*). This is supported by the findings that mutations in the S4-S5 linker (N225-K228) do not significantly affect the coupling between VSD and the pore (*Sun and Horrigan, 2022*). Arguably, the distinct features of voltage-sensing and pore-sensor coupling observed in BK channels are intimately related to the novel non-domain-swapped TM topology. In particular, the VSD and pore domains are more tightly packed compared to the domain-swapped configuration (*Figure 1*), which restricts the VSD motions and provides a direct pathway for their inter-talk. Interestingly, a similar non-canonical pathway for VSD-pore coupling, involving the S4/S1 and S1/S5 interfaces, has recently been proposed for the non-domain-swapped cardiac hERG potassium channel (*Bassetto et al., 2023*). We also note that a similar non-canonical VSD-pore coupling involving S4-S5 interactions between neighboring subunits has been suggested to complement the canonical coupling mode even in domain-swapped Kv channels (*Hou et al., 2020*; *Fernández-Mariño et al., 2018*; *Carvalho-de-Souza and Bezanilla, 2019*). An increasing number of ion channels have been discovered to adopt non-domain-swapped TM topology besides BK, including hERG (*Wang and MacKinnon, 2017*), KvAP (*Tao and MacKinnon, 2019b*), HCN (*Lee and MacKinnon, 2017*), Eag1 (*Whicher and MacKinnon, 2016*), and CNG channels (*Li et al., 2017*). It is possible that the mechanistic features observed for BK channels may apply to voltage gating of the important emerging class of non-domain swapped ion channels in general.

Importantly, full-length BK channels can be independently activated by membrane potential and intracellular $Ca^{2+}$ (*Horrigan and Aldrich, 2002*; *Shi and Cui, 2001*; *Gessner et al., 2012*). The latter requires the C-terminal gating ring that is absent in the Core-MT construct studied in this work. Cryo-EM structures have revealed functional state-dependent interactions between the gating ring and VSD (*Hite et al., 2017*; *Tao et al., 2017*; *Tao and MacKinnon, 2019a*; *Gustavo et al., 2023*), strongly supporting that the calcium sensor and VSD are coupled. It is not clear how the VSD-gating ring coupling may affect the nature of VSD movements or how VSDs may drive the pore opening. Nonetheless, mechanistic details revealed from the voltage-gating of the Core-MT construct should provide a solid basis for future computational and experimental studies of BK activation and regulation.

## Materials and methods
### Molecular modeling and atomistic simulations

The structure of the Core-MT human BK channel in the closed state was derived from the Cryo-EM structures of the *ac*BK channel in deactivated $Ca^{2+}$-free states [PDB 5tji (*Tao et al., 2017*)] as previously described (*Jia et al., 2018*; *Yazdani et al., 2020*). The simulated construct was truncated at R342. The 11-residue C-terminal Kv mini-tails are not involved in voltage gating (*Budelli et al., 2013*; *Zhang et al., 2017*), and they were thus not included. The dynamic loop (C54-V91) and N-terminal tail (M1-N19) were not included either. Residues before and after the missing segments are capped with either an acetyl group (for N-terminus) or a N-methyl amide (for C-terminus). Standard protonation states under neutral pH were assigned for all titratable residues.

The initial structures were first inserted in model POPC lipid bilayers and then solvated in TIP3P water using the CHARMM-GUI web server (*Lee et al., 2016*). Even though polarizable water models are probably necessary to capture the precise energetics (and kinetics) of dewetting transitions within a protein cavity, it has also been shown that classical nonpolarizable water models such as TIP3P are sufficient to capture the spontaneous dewetting and rehydration of BK channels (*Jia et al., 2018*; *Gu and de Groot, 2023*; *Nordquist et al., 2023*; *Coronel et al., 2024*). In fact, a recent analysis using the Drude polarizable force field actually under-estimated the hydration free energy of deactivated BK pore (~0.5 kcal/mol) (*Deng and Cui, 2024*), which is very likely too small to sustain a dry and nonconductive pore. All systems were neutralized and 150 mM KCl was added. The final simulation boxes contain about 593 lipid molecules (POPC) and ~50,000 water molecules and other solutes, with a total of ~250,000 atoms and dimensions of ~160 × 160×110 Å$^3$. The CHARMM36m all-atom force field (*Huang et al., 2017*) and the CHARMM36 lipid force field (*Klauda et al., 2010*) were used. All simulations were performed using Desmond (*Bowe et al., 2006*) on Anton 2 (*Shaw et al., 2008*; *Shaw et al., 2014*) or CUDA-enabled versions of Gromacs 2020 (*Hess et al., 2008*; *Abraham et al.,*

*2015*) on GPU clusters. Electrostatic interactions were described by using the Particle Mesh Ewald (PME) algorithm (*Darden et al., 1993*) with a cutoff of 12 Å. Van der Waals interactions were cutoff at 12 Å with a smooth switching function starting at 10 Å. Covalent bonds to hydrogen atoms were constrained by the SHAKE algorithm (*Ryckaert et al., 1977*), and the MD time step was set at 2 fs. The temperature was maintained at 298 K using the Nose-Hoover thermostat (*Nosé, 1984*; *Hoover, 1985*) (in Gromacs). The pressure was maintained semi-isotopically at 1 bar at membrane lateral directions using the Parrinello–Rahman barostat algorithm (*Parrinello and Rahman, 1981*).

All systems were first minimized for 5000 steps using the steepest descent algorithm, followed by a series of equilibration steps where the positions of heavy atoms of the protein and lipid were harmonically restrained as prescribed by the CHARMM-GUI Membrane Builder (*Jo et al., 2007*). For the Anton simulation, an additional 70 ns equilibration step with 0.1 kcal.mol⁻¹.Å⁻² position restraints on all protein heavy atoms was performed before production runs. Note that the inner pore became dewetted during equilibration simulations. All production simulations were performed under NVT (constant particle number, volume, and temperature) conditions at 298 K. The P-loop/filter (T273 to D292) and C-terminus were harmonically restrained with a small force constant of 0.1 kcal.mol⁻¹.Å⁻² in all production simulations to prevent drift in all production simulations. Using Anton 2, a 2-μs control simulation was first performed without membrane voltage (*sim 1*, *Supplementary file 1*). Two independent 10-μs simulations were then performed at 750 mV to directly probe voltage-driven activation transitions (*sim 2*). The voltage was applied as a constant external electric field ($E=V/L_z$) imposed along the z dimension (*Roux, 2008*).

## Steered molecular dynamics simulations

Steered molecular dynamics (SMD) simulations were performed to test the role of R210 and R213 in voltage sensing and channel activation. In these simulations, a moving reference point was used to simulate the effect of an external electric field by pulling the CZ atoms of R210 and R213 from their initial positions in the resting state to those observed in the activated state. As illustrated in *Figure 4—figure supplement 1*, the initial reference points of SMD were positioned at the location of each CZ atom of R210 and R213 in the resting/closed state. From 0 to 100 ns, these reference points were moved along the z-axis at constant speed until they reached the positions corresponding to the predicted activated VSD state at 100 ns. A harmonic positional restraint of 5 kcal/mol·Å² was applied between each CZ atom to its respective reference point along z-axis, allowing the CZ atom to track the reference point's movement. After 100 ns, the reference points were held stationary at the positions corresponding to the activated VSD state, with the same harmonic restraints imposed to maintain the CZ atoms in their activated position. Four independent simulations (*sim 8*) were performed, and pore opening was observed in two of the four replicates (replicas 2 and 4). In the other two replicas, the S5-S6 packing was broken during the first 100 ns, likely due to strong steering potentials imposed, and the simulations were terminated.

## Free energy analysis of gating charges

The total gating charge ΔQ is the sum of contributions of all charged residues in VSD and can be calculated as (*Khalili-Araghi et al., 2010*; *Roux, 2008*):

$$\Delta Q = \sum_i q_i \left[ f_c \left( i \right) - f_o \left( i \right) \right], \tag{1}$$

where $q_i$ is the charge of a specific residue, and the dimensionless quantities $f_s(i)$ for a specific conformational state s, which can be open (o) or closed (c), represent the coupling of charge $q_i$ to the transmembrane potential. The value of $f_s(i)$ is derived from the difference between charging free energies calculated at two different voltages $V_1$ and $V2$,

$$f_s \left( i \right) = \frac{\Delta G_s \left( V_2, q_i \right) - \Delta G_s \left( V_1, q_i \right)}{q_i \left( V_2 - V_1 \right)}, \tag{2}$$

where $\Delta G_s \left( V, q_i \right)$ is the free energy cost of increasing the charge of residue i from 0 to $q_i$ in state s under membrane voltage V. $\Delta G_s \left( V, q_i \right)$ was calculated using thermodynamic integration (TI), with the charge gradually scaled to the final value in increments of 0.1, that is $\lambda$=0, 0.1, 0.2, ...., 1.0. In this

work, we focused on seven charged residues on VSD, namely, D153, R167, D186, R207, R210, R213, and E219, and performed four TI free energy calculations for each residue (sim 3–6, *Supplementary file 1*). The last snapshot of simulation 2b was used to represent the activated state of VSD. For each TI window, the system was first equilibrated for 200 ps and then simulated for 4 ns to calculate the mean force $<\delta H/\delta\lambda>_\lambda$.

## Structural and electrostatic potential analysis

The displacement of residue charged groups, as well as movement of helices (translocation, titling, and rotation) is calculated using MDanalysis (*Michaud-Agrawal et al., 2011*) together with in-house scripts. The TM helices are defined as: T109-S135 (S1), F148-A170 (S2), V181-L199 (S3), G205-N225 (S4), S230-S259 (S5), and A313-E324 (S6 after the glycine hinge). Residue contacts were identified using a minimal heavy atom distance cutoff of 5 Å. Pore water molecules were identified as those occupying the inner pore cavity below the selectivity filter, roughly from L312 to the plane defined by the center of mass of P320. Pore profiles are calculated using program HOLE (*Smart et al., 1996*). To examine how the protein modulates the local electric field, average electrostatic potential maps were then calculated from the first (resting) and last (activated) 250 ns of *sim 2b* using a combination of Python scripts and the VMD's PMEPot and VolMap plugins (*Aksimentiev and Schulten, 2005*). The electrostatic potential is obtained by solving the Poisson equation $\nabla^2\varnothing\left(r\right) = -4\pi\left(\sum_i \rho_i\left(r\right)\right)$ where the sum runs over all atoms, and $\rho_i(r)$ is the charge electric potential distribution contributed by atom $i$ at position $r$ approximated by a spherical Gaussian: $\rho_i\left(r\right) = q_i\left(\frac{\beta}{\sqrt{\pi}}\right)^3 e^{-\beta^2\left(r-r_i\right)^2}$. An Ewald factor $\beta$ of 0.25 Å$^{-1}$ was used for the inverse width of the Gaussian and grid size was set to 1 Å. All molecular illustrations were prepared using VMD (*Humphrey et al., 1996*). External electric potential was added to the PMEPot grid file with in-house scripts (see *Supplementary file 1*).

## Dynamic community and coupling pathways

Dynamic network analysis was performed using the *Networkview* (*Eargle and Luthey-Schulten, 2012*) plugin of VMD. To build the network, each amino acid was represented as a single node at the Cα position, and a contact (edge) was defined between two nodes if the minimal heavy-atom distance between residues was within a cutoff distance (5 Å) during at least 75% of the trajectory. The resulting contact matrix was weighted based on the covariance of dynamic fluctuation ($C_{ij}$) calculated from the same MD trajectory as $w_{ij}$ = - log($|C_{ij}|$). The length of a possible pathway $D_{ij}$ between distant nodes $i$ and $j$ is defined as the sum of the edge weights between consecutive nodes along this path. The shortest path, calculated using the Floyd-Warshall algorithm (*Floyd, 1962*), is considered the optimal pathway with the strongest dynamic coupling. Suboptimal paths are identified as alternative top-ranked paths with lengths that deviate by less than 50% from the optimal path. We further performed dynamic community analysis to identify groups of residues that are more tightly coupled among themselves (*Girvan and Newman, 2002*).

## Information flow analysis

Information flow provides a global assessment of the contributions of all nodes to the dynamic coupling between selected "source" and "sink" nodes (*Kang et al., 2020*; *Westerlund et al., 2020*). This analysis complements the dynamic pathway analysis to provide additional insights on how different residues may contribute to sensor-pore coupling. For this, a network similar to the one described above was first constructed. Pairwise mutual information was calculated between node $i$ and $j$ as follows: $M_{ij} = H_i + _{Hj} - H_{ij}$. $H_i$ is calculated as $\frac{1}{N}\sum_{n=1}^{N}[-ln\,\rho_i(x)]$, where $\rho_i(x)$ is the fluctuation density and $x$ is the distance to the equilibrium position (*Westerlund et al., 2020*). Gaussian mixture model (GMM; *Dempster et al., 1977*) is used to estimate the density. The residue network is then defined as $A_{ij} = C_{ij} M_{ij}$, where $C_{ij}$ is the contact map. To analyze the information flow from the source ($S_0$) to sink ($S_l$) nodes, the network Laplacian is calculated as $L = D$ A, where D is diagonal degree matrix: $D_{ii} = \sum_j A_{ij}$. The information flow through a given node (residue) is defined as $f_i = \frac{1}{2}\left(\sum_j \left|(P_i - P_j)\right| A_{ij}\right)$. The potential $P$ is given by $P = \widetilde{L}^{-1} b$, where $\widetilde{L}^{-1}$ is the inverse reduced Laplacian, and $b$ is the supply vector that corresponds to one unit of current entering at the source node that will exit at sink nodes. The magnitude of $f_i$ thus quantifies the contribution of residue $i$ to dynamic coupling between the source and sink nodes.

## Acknowledgements

The authors thank Jianmin Cui and Guohui Zhang for critical discussions. This work is supported by NIH R35 GM144045 (Chen). Anton 2 computer time was provided by the Pittsburgh Supercomputing Center (PSC) through Grant R01 GM116961 from the National Institutes of Health. The Anton 2 machine at PSC was generously made available by DE Shaw Research.

## Additional information

### Funding

| Funder | Grant reference number | Author |
| --- | --- | --- |
| National Institute of General Medical Sciences | R35 GM144045 | Jianhan Chen |
| National Institute of General Medical Sciences | R01 GM116961 | Zhiguang Jia<br>Jianhan Chen |

The funders had no role in study design, data collection and interpretation, or the decision to submit the work for publication.

### Author contributions

Zhiguang Jia, Conceptualization, Formal analysis, Investigation, Visualization, Writing – original draft, Writing – review and editing; Jianhan Chen, Conceptualization, Formal analysis, Funding acquisition, Writing – original draft, Project administration, Writing – review and editing

### Author ORCIDs

Jianhan Chen ⓘ https://orcid.org/0000-0002-5281-1150

Reviewer #1 (Public review): https://doi.org/10.7554/eLife.105895.4.sa1
Reviewer #2 (Public review): https://doi.org/10.7554/eLife.105895.4.sa2
Author response https://doi.org/10.7554/eLife.105895.4.sa3

## Additional files

### Supplementary files

Supplementary file 1. Summary of simulations.

MDAR checklist

### Data availability

All data needed to evaluate the conclusions in the paper are present in the paper and the Supplementary Materials. The PDB structures of Core-MT with resting and activated VSDs, as well as simulation input and analysis script, can be found on GitHub at: https://github.com/mdlab-um/Votage_gating_Core-MT (copy archived at *mdlab-um, 2024*).

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
