## [Editor Report · eLife Assessment]

This **valuable** study addresses the structural basis of voltage-activation of BK channels using atomistic simulations of several microseconds, to assess conformational changes that underlie both voltage-sensing and gating of the pore. The findings, including movement of specific charged residues, combined with the degree to which these movements are coupled to pore movements, provide a **solid** basis for understanding voltage-gating mechanisms in this class of channels. This paper will likely be of interest to ion channel biologists and biophysicists focused on voltage-dependent channel gating mechanisms.

---

## [Referee Report · Reviewer #1 (Public review)]

Summary:

This study provides new insight into the non-canonicial voltage-gating mechanism of BK channels through prolonged (10 us) MD simulations of the Slo1 transmembrane domain conformation and K+ conduction in response to high imposed voltages (300, 750 mV). The results support previous conclusions based on functional and structural data and MD simulations that the voltage-sensor domain (VSD) of Slo1 undergoes limited conformational changes compared to Kv channels, and predicts gating charge movement comparable in magnitude to experimental results. The gating charge calculations further indicate that R213 and R210 in S4 are the main contributors owing to their large side chain movements and the presence of a locally focused electric field, consistent with recent experimental and MD simulation results by Carrasquel-Ursulaez et al.,2022. Most interestingly, changes in pore conformation and K+ conduction driven by VSD activation are resolved, providing information regarding changes in VSD/pore interaction through S4/S5/S6 segments proposed to underly electromechanical coupling.

Strengths:

Include that the prolonged timescale and high voltage of the simulation allow apparent equilibration in the voltage-sensor domain (VSD) conformational changes and at least partial opening of the pore. The study extends the results of previous MD simulations of VSD activation by providing quantitative estimates of gating charge movement, showing how the electric field distribution across the VSD is altered in resting and activated states, and testing the hypothesis that R213 and R210 are the primary gating charges by steered MD simulations. The ability to estimate gating charge contributions of individual residues in the WT channel is useful as a comparison to experimental studies based on mutagenesis which have yielded conflicting results that could reflect perturbations in structure. Use of dynamic community analysis to identify coupling pathways and information flow for VSD-pore (electromechanical) coupling as well as analysis of state-dependent S4/S5/S6 interactions that could mediate coupling provide useful predictions extending beyond what has been experimentally tested.

Weaknesses:

Weaknesses include that a truncated channel (lacking the C-terminal gating ring) was used for simulations, which is known to have reduced single channel conductance and electromechanical coupling compared to the full-length channel. In addition, as VSD activation in BK channels is much faster than opening, the timescale of simulations was likely insufficient to achieve a fully open state as supported by differences in the degree of pore expansion in replicate simulations, which are also smaller than observed in Ca-bound open structures of the full-length channel. Taken together, these limitations suggest that inferences regarding coupling pathways and interactions in the fully open voltage-activated channel may be only partially supported and therefore incomplete. That said, adequate discussion regarding these limitations are provided together with dynamic community analysis based on the Ca-bound open structure. The latter supports the main conclusions based on simulations, while providing an indication of potential interaction differences between simulated and fully open conformations. Another limitation is that while the simulations convincingly demonstrate voltage-dependent channel opening as evidenced by pore expansion and conduction of K+ and water through the pore, single channel conductance is underestimated by at least an order of magnitude, as in previous studies of other K+ channels. These quantitative discrepancies suggest that MD simulations may not yet be sufficiently advanced to provide insight into mechanisms underlying the extraordinarily large conductance of BK channels.

Comments on revisions:

My previous questions and concerns have been adequately addressed.

---

## [Referee Report · Reviewer #2 (Public review)]

Summary:

The manuscript by Jia and Chen addresses the structural basis of voltage-activation of BK channels using computational approaches. Although a number of experimental studies using gating current and patch-clamp recording have analyzed voltage-activation in terms of observed charge movements and the apparent energetic coupling between voltage-sensor movement and channel opening, the structural changes that underlie this phenomenon have been unclear. The present studies use a reduced molecular system comprising the transmembrane portion of the BK channel (i.e. the cytosolic domain was deleted), embedded in a POPC membrane, with either 0 or 750 mV applied across the membrane. This system enabled acquisition of long simulations of 10 microseconds, to permit tracking of conformational changes of the channel. The authors principal findings were that the side chains of R210 and R213 rapidly moved toward the extracellular side of the membrane (by 8 - 10 Å), with greater displacements than any of the other charged transmembrane residues. These movements appeared tightly coupled to movement of the pore-lining helix, pore hydration, and ion permeation. The authors estimate that R210 and R213 contribute 0.25 and 0.19 elementary charges per residue to the gating current, which is roughly consistent with estimates based on electrophysiological measurements that used the full-length channel.

Strengths:

The methodologies used in this work are sound, and these studies certainly contribute to our understanding of voltage-gating of BK channels. An intriguing observation is the strongly coupled movement of the S4, S5, and S6 helices that appear to underlie voltage-dependent opening. Based on Fig 2a-d, the substantial movements of the R210 and R213 side chains occur nearly simultaneously to the S6 movement (between 4 - 5 usec of simulation time). This seems to provide support for a "helix-packing" mechanism of voltage gating in the so-called "non-domain-swapped" voltage-gated K channels.

Weaknesses:

The main limitation is that these studies used a truncated version of the BK channel, and there are likely to be differences in VSD-pore coupling in the context of the full-length channels that will not be resolved in the present work. Nonetheless, the authors provide a strong rationale for their use of the truncated channel, and the results presented will provide a good starting point for future computational studies of this channel.

---

## [Author Response]

The following is the authors’ response to the previous reviews

**Recommendations for the authors:**

**Reviewing Editor Comments:**
The resubmitted version of the manuscript adequately addressed several initial comments made by reviewing editors, including a more detailed analysis of the results (such as those of bilayer thickness). This version was seen by 2 reviewers. Both reviewers recognize this work as being an important contribution to the field of BK and voltage-dependent ion channels in general. The long trajectories and the rigorous/novel analyses have revealed important insights into the mechanisms of voltage-sensing and electromechanical coupling in the context of a truncated variant of the BK channel. Many of these observations are consistent with structural and functional measurements of the channel, available thus far. The authors also identify a novel partially expanded state of the channel pore that is accessed after gating-charge displacement, which informs the sequence of structural events accompanying voltage-dependent opening of BK.However, there are key concerns regarding the use of the truncated channel in the simulations. While many gating features of BK are preserved in the truncated variant, studies have suggested that opening of the channel pore to voltage-sensing domain rearrangement is impaired upon gating-ring deletion. So the inferences made here might only represent a partial view of the mechanism of electromechanical coupling.It is also not entirely clear whether the partially expanded pore represents a functionally open, sub-conductance, or another closed state. Although the authors provide evidence that the inner pore is hydrated in this partially open state, in the absence of additional structural/functional restraints, a confident assignment of a functional state to this structure state is difficult. Functional measurements of the truncated channel seem to suggest that not only is their single channel conductance lower than full-length channels, but they also appear to have a voltage-independent step that causes the gates to open. It is unclear whether it is this voltage-independent step that remains to be captured in these MD trajectories. A clean cut resolution of this conundrum might not be feasible at this time, but it could help present the various possibilities to the readers.

We appreciate the positive comments and agree that there will likely be important differences between the mechanistic details of voltage activation between the Core-MT and full-length constructs of BK channels. We also agree that the dilated pore observed in the simulation may not be the fully open state of Core-MT.

Nonetheless, the notion that the simulation may not have captured the full pore opening transition or the contribution of the CTD should not render the current work “incomplete”, because a complete understanding of BK activation would be an unrealistic goal beyond the scope of this work. We respectfully emphasize that the main insights of the current simulations are the mechanisms of voltage sensing (e.g., the nature of VSD movements, contributions of various charged residues, how small charge movements allow voltage sensing, etc.) as well as the role of the S4-S5-S6 interface in VSD-pore coupling. As noted by the Editor and reviewers, these insights represent important steps towards establishing a more complete understanding of BK activation.

Below are the specific comments of the two experts who have assessed the work and made specific suggestions to improve the manuscript.

**Reviewer #1 (Recommendations for the authors):**
(1) Although the successful simulation of V-dependent K+ conduction through the BK channel pore and analysis of associated state dependent VSD/pore interactions and coupling analysis is significant, there are two related questions that are relevant to the conclusions and of interest to the BK channel community which I think should be addressed or discussed.One key feature of BK channels is their extraordinarily large conductance compared to other K+ selective channels. Do the simulations of K+ conductance provide any insight into this difference? Is the predicted conductance of BK larger than that of other K+ channels studied by similar methods? Is there any difference in the conductance mechanism (e.g., the hard and soft knock-on effects mentioned for BK)?

The molecular basis of the large conductance of BK channels is indeed an interesting and fundamental question. Unfortunately, this is beyond the scope of this work and the current simulation does not appear to provide any insight into the basis of large conductance. It is interesting to note, though, the conductance is apparently related to the level of pore dilation and the pore hydration level, as increasing hydration level from ~30 to ~40 waters in the pore increases the simulated conductance from ~1.5 to 6 pS (page 8). This is consistent with previous atomistic simulations (Gu and de Groot, Nature Communications 2023; ref. 33) showing that the pore hydration level is strongly correlated with observed conductance. As noted in the manuscript, the conductance mechanism through the filter appears highly similar to previous simulations of other K+ channels (Page 8). Given the limit conductance events observed in the current simulations, we will refrain from discussing possible basis of the large conductance in BK channels except commenting on the role of pore hydration (page 8; also see below in response to #5).

The pore in the MD simulations does not open as wide as the Ca-bound open structure, which (as the authors note) may mean that full opening requires longer than 10 us. I think that is highly likely given that the two 750 mV simulations yielded different degrees of opening and that in BK channels opening is generally much slower than charge movement. Therefore, a question is - do any of the conclusions illustrated in Figures 6, S5, S6 differ if the Ca-bound structure is used as the open state? For example, I expect the interactions between S5 and S6 might at least change to some extent as S6 moves to its final position. In this case, would conclusions about which residues interact, and get stronger or weaker, be the same as in Figures S6 b,c? Providing a comparison may help indicate to what extent the conclusions are dependent on achieving a fully open conformation.

We appreciate the reviewer’s suggestion and have further analyzed the information flow and coupling pathways using the simulation trajectory initiated from the Ca^2+^-bound cryo-EM structure (sim 7, Table S1). The new results are shown in two new SI Figures S7 and S8, and new discussion has been added to pages 14-15. Comparing Figures 5 and S7, we find that dynamic community, coupling pathways, and information flow are highly similar between simulation of the open and closed states, even though there are significant differences in S5 contacts in the simulated open state vs Ca^2+^-bound open state (Figure S8). Interestingly, there are significant differences in S4-S5 packing in the simulated and Ca^2+^-bound open states (Figure S8 top panel), which likely reflect important difference in VSD/pore interactions during voltage vs Ca^2+^ activation.

(2) P4 Significance -"first, successful direct simulation of voltage-activation"This statement may need rewording. As noted above Carrasquel-Ursulaez et al.,2022 (reference 39) simulated voltage sensor activation under comparable conditions to the current manuscript (3.9 us simulation at +400 mV), and made some similar conclusions regarding R210, R213 movement, and electric field focusing within the VSD. However, they did not report what happens to the pore or simulate K+ movement. So do the authors here mean something like "first, successful direct simulation of voltage-dependent channel opening"?

We agree with the reviewer and have revised the statement to “ … the first successful direct simulation of voltage-dependent activation of the big potassium (BK) channel, ..”

(3) P5 "We compare the membrane thickness at 300 and 750 mV and the results reveal no significant difference in the membrane thickness (Figure S2)"The figure also shows membrane thickness at 0 mV and indicates it is 1.4 Angstroms less than that at 300 or 750 mV. Whether or not this difference is significant should be stated, as the question being addressed is whether the structure is perturbed owing to the use of non-physiological voltages (which would include both 300 and 750 mV).

We have revised the Figure S2 caption to clarify that one-way ANOVA suggest the difference is not significant.

(4) P7 "It should be noted that the full-length BK channel in the Ca2+ bound state has an even larger intracellular opening (Figure 2f, green trace), suggesting that additional dilation of the pore mayoccur at longer timescales."As noted above, I agree it is likely that additional pore dilation may occur at longer timescales. However, for completeness, I suppose an alternative hypothesis should be noted, e.g. "...suggesting that additional dilation of the pore may occur at longer timescales, or in response to Ca-binding to the full length channel."

This is a great suggestion. Revised as suggested.

(5) Since the authors raise the possibility that they are simulating a subconductance state, some more discussion on this point would be helpful, especially in relation to the hydrophobic gate concept. Although the Magleby group concluded that the cytoplasmic mouth of the (fully open) pore has little impact on single channel conductance, that doesn't rule out that it becomes limiting in a partially open conformation. The simulation in Figure 3A shows an initial hydration of the pore with ~15 waters with little conductance events, suggesting that hydration per se may not suffice to define a fully open state. Indeed, the authors indicate that the simulated open state (w/ ~30-40 waters) has 1/4th the simulated conductance of the open structure (w/ ~60 waters). So is it the degree of hydration that limits conductance? Or is there a threshold of hydration that permits conductance and then other factors that limit conductance until the pore widens further? Addressing these issues might also be relevant to understanding the extraordinarily large conductance of fully open BK compared to other K channels.

We agree with the reviewer’s proposal that pore hydration seems to be a major factor that can affect conductance. This is also well in-line with the previous computational study by Gu and de Groot (2023). We have now added a brief discussion on page 8, stating “Besides the limitation of the current fixed charge force fields in quantitively predicting channel conductance, we note that the molecular basis for the large conductance of BK channels is actually poorly understood (78). It is noteworthy that the pore hydration level appears to be an important factor in determining the apparent conductance in the simulation, which has also been proposed in a previous atomistic simulation study of the Aplysia BK channel (33).”

Minor points(1) P5 "the fully relaxed pore profile (red trace in Figure S1d, top row) shows substantial differences compared to that of the Ca2+-free Cryo-EM structure of the full-length channel."For clarity, I suggest indicating which is the Ca-free profile - "... Ca2+-free Cryo-EM structure of the full-length channel (black trace)."

We greatly appreciate the thoughtful suggestion. Revised as suggested.

(2) P8 "Consistent with previous simulations (78-80), the conductance follows a multi-ion mechanism, where there are at least two K+ ions inside the filter"For clarity, I suggest indicating these are not previous simulations of BK channels (e.g., "previous simulations of other K+ channels ...").

Author response: Revised as suggested. Thank you.

(3) Figure 2, S1 - grey traces representing individual subunits are very difficult to see (especially if printed). I wonder if they should be made slightly darker. Similar traces in Figure 3 are easier to see.

The traces in Figure S1 are actually the same thickness in Figure 3 and they appear lighter due to the size of the figure. Figure 2 panels a-c have been updated to improve the resolution.

(4) Figure 2 - suggest labeling S6 as "S6 313-324" (similar to S4 notation) to indicate it is not the entire segment.

Figure 2 panel (d) has been updated as suggested.

(5) Figure 2 legend - "Voltage activation of Core-MT BK channels. (a-d)..."It would be easier to find details corresponding to individual panels if they were referenced individually. For example:"(a-d) results from a 10-μs simulation under 750 mV (sim2b in Table S1). Each data point represents the average of four subunits for a given snapshot (thin grey lines), and the colored thick lines plot the running average. (a) z-displacement of key side chain charged groups from initial positions. The locations of charged groups were taken as those of guanidinium CZ atoms (for Arg) and sidechain carboxyl carbons (for Asp/Glu) (b) z-displacement of centers-of-mass of VSD helices from initial positions, (c) backbone RMSD of the pore-lining S6 (F307-L325) to the open state, and (d) tilt angles of all TM helices. Only residues 313-324 of S6 were included inthe tilt angle calculation, and the values in the open and closed Cryo-EM structures are marked using purple dashed lines. "

We appreciate the thoughtful suggestion and have revised the caption as suggested.

(6) Figure S1 - column labels a,b,c, and d should be referenced in the legend.

The references to column labels have been added to Figure S1 caption.

(7) References need to be double-checked for duplicates and formatting.a) I noticed several duplicate references, but did not do a complete search: Budelli et al 2013 (#68, 100), Horrigan Aldrich 2002 (#22,97), Sun Horrigan 2022 (#40, 86), Jensen et al 2012 (#56,81).b) Reference #38 is incorrectly cited with the first name spelled out and the last name abbreviated.

We appreciate the careful proofreading of the reviewer. The duplicated references were introduced by mistake due to the use of multiple reference libraries. We have gone through the manuscript and removed a total of 5 duplicated references.

Response to additional reviewer commentsMy only new comment is that the numbering of residues in Fig. S8 does not match the standard convention for hSlo and needs to be doublechecked. For the residues I checked, the numbers appear to be shifted 3 compared hSlo (e.g. Y315, P317, E318, G324 should be Y318, P320, E321, G327).

We greatly appreciate the reviewer for catching the errors in residue labels. Figure S8 has now been updated to include correct residue labels. Thanks!

**Reviewer #2 (Recommendations for the authors):**
This manuscript has been through a previous level of review. The authors have provided their responses to the previous reviewers, which appear to be satisfactory, and I have no additional comments, beyond the caveats concerning interpretations based on the truncated channel, which are noted above.

We greatly appreciate the constructive comments and insightful advice. Please see above response to the Reviewing Editor’s comments for response and changes regarding the caveats concerning interpretations of the current simulations.